

**Fire-regime variability impacts forest carbon dynamics for centuries to millennia**
**Tara W. Hudiburg[1], Philip E. Higuera[2], and Jeffrey A. Hicke[3]**
[1]Department of Forest, Rangeland, and Fire Sciences
University of Idaho
875 Perimeter Dr.
Moscow, ID 83844-1133
[2]Department of Ecosystem and Conservation Sciences
University of Montana
32 Campus Dr.
Missoula, MT 59812
[3]Department of Geography
University of Idaho
875 Perimeter Dr.
Moscow, ID 83844-3021
**\*Corresponding Author**
thudiburg@uidaho.edu
phone: 208-885-7044
fax: 208-885-5534
*Running header:*
Fire-regime variability impacts on forest carbon
*Keywords:*
Fire regimes, forest carbon, paleoecology, ecosystem modeling, Rocky Mountains, Rocky
Mountain National Park, lodgepole pine
*Type of paper:*
Primary research article



**Abstract**

Wildfire is a dominant disturbance agent in forest ecosystems, shaping important biogeochemical processes including net carbon (C) balance. Long-term monitoring and chronosequence studies highlight a resilience of biogeochemical properties to large, stand-replacing, high-severity fire events. In contrast, the consequences of repeated fires or temporal variability in a fire regime (e.g., the characteristic timing or severity of fire) are largely unknown, yet theory suggests that such variability could strongly influence forest C trajectories for millennia. Here we combine a 4500-year paleoecological record of fire activity with ecosystem modeling to investigate how fire-regime variability impacts soil C and net ecosystem carbon balance. We found that C trajectories in a paleo-informed scenario differed significantly from an equilibrium scenario (with a constant fire return interval), largely due to variability in the timing and severity of past fires. Paleo-informed scenarios contained multi-century periods of positive and negative net ecosystem C balance, with magnitudes significantly larger than observed under the equilibrium scenario. Further, this variability created legacies in soil C trajectories that lasted for millennia, and was of a magnitude great than simulated under an equilibrium, climate-warming scenario (i.e., 2 °C growing season warming). Our results imply that fire-regime variability is a major driver of C trajectories in stand-replacing fire regimes. Predicting carbon balance in these systems, therefore, will depend strongly on the ability of ecosystem models to represent a realistic range of fire-regime variability over the past several centuries to millennia.



## 1. Introduction

Wildfire is a pervasive disturbance agent in forest ecosystems, strongly shaping ecosystem structure and function, including vegetation composition, nutrient cycling, and energy flow. While the immediate impacts of disturbance can be dramatic, the longevity of these impacts is less clear. In ecosystems where disturbance is historically prevalent, vegetation and biogeochemical properties typically return to pre-disturbance conditions over years to decades (Dunnette et al., 2014; McLauchlan et al., 2014), motivating the concept of "biogeochemical resilience" (Smithwick, 2011). Characterizing biogeochemical resilience emphasizes understanding pool sizes and changes to inputs or outputs of key elements (McLauchlan et al., 2014; Smithwick, 2011). In the context of wildfire, biogeochemical resilience is determined by pool sizes prior to a fire event, elemental losses and transformations that occur during and shortly after a fire event (e.g., from volatilization and erosion), and post-fire changes in elemental pools, which in turn are determined by the rate and composition of post-fire revegetation (McLauchlan et al., 2014; Schlesinger et al., 2015; Smithwick, 2011).

Changes in the characteristic frequency or severity of fire (i.e., the fire regime) are therefore predicted to lead to compounding and potentially long-lasting changes or shifts in biogeochemical states. For example, increased disturbance frequency can deplete key growth-limiting nutrients (Yelenik et al., 2013), potentially influencing ecosystem trajectories for decades to centuries (McLauchlan et al., 2014). Net ecosystem carbon balance (NECB) is also highly sensitive to disturbance (Hudiburg et al., 2011), and while NECB trends towards 0 under a uniform disturbance regime (Chapin et al., 2006), shifting disturbance regimes may alter NECB over centuries to millennia (Goetz et al., 2012; Kelly et al., 2016). While these ideas have a strong conceptual basis and empirical support on decadal timescales, we have lacked the data needed to test them over longer timescales – and to consider their implications for future projections – until only recently.

Coupling paleo observations (i.e. "paleo-informed") with ecosystem modeling provides an important tool for assessing the impacts of fire-regime variability on biogeochemical dynamics by combining the mechanistic representation of ecosystem processes with actual patterns of fire activity reconstructed from the past. For example, in Alaskan boreal forests paleo-informed ecosystem modeling highlights fire as the dominant control on C cycling over the past





millennium, far outweighing the effects of climate variability (Kelly et al., 2016). Given the
significance influence of fire, estimates of modern C states ("initial conditions" for modeling
future C states) can be highly sensitive to assumptions about the past fire activity. Ecosystem
models typically require a 'spin up' period to equilibrate C and N pools and can include a fixed
disturbance interval (e.g., a constant fire return interval), resulting in ecosystem C and N
trajectories that are in 'equilibrium' with climate, ecosystem properties, and the disturbance
regime. Following centuries of equilibrium, known disturbance events from the historical record
are included, and the final results are used for initial conditions (baseline) for future scenarios.
However, paleo-informed disturbance histories spanning many centuries can result in initial
conditions that differ from equilibrium runs. In the boreal example, forests were a small net C
source over the past several decades in paleo-informed simulations, whereas forests were a small
net C sink when a constant fire return interval was assumed (Kelly et al., 2016). We would
expect a similar sensitivity of C dynamics to fire in other stand-replacing fire regimes, although
specific trajectories and impacts on modern states could vary widely, contingent on the specific
history of fire activity.
Here, we pair a paleoecological record of vegetation and wildfire activity in a subalpine forest
(Dunnette et al., 2014) with an ecosystem model to evaluate the sensitivity of forest ecosystem
processes to fire-regime variability over a 4500-year period. Our paleoecological record reveals
the timing and severity of past wildfire activity within a subalpine forest watershed that was
consistently dominated by lodgepole pine (*Pinus contorta*). We use this record to drive fire
disturbances in an ecosystem model and test alternative hypotheses that help reveal the potential
patterns and mechanisms causing past ecosystem change, focusing on a slowly varying carbon
pool (soil C) and net ecosystem carbon balance (NECB). The resulting trends provide theoretical
insight into how observed fire-regime variability can affect carbon trajectories from decadal to
millennial scales. Through a series of paleo-informed and control modeling scenarios, we
address three key questions about the biogeochemical impacts and legacies of wildfire activity:
(1) how does centennial-to-millennial-scale variability in fire activity impact biogeochemical
processes that regulate soil C and NECB; (2) for how long does the legacy wildfire activity
impact current ecosystem states; and (3) what is the magnitude of these impacts relative to the
impacts of climatic warming. Our results highlight the importance of fire activity in shaping



ecosystem C dynamics across a range of time scales, and they have important implications for
projecting future ecosystem states under scenarios of climate and disturbance-regime change.

## 2    Materials and Methods

### 2.1 Study sites

We studied the biogeochemical consequences of fire-regime variability by informing the
DayCent model with fire history data derived from sedimentary charcoal preserved in Chickaree
Lake, Colorado (Dunnette et al., 2014). Chickaree Lake (40.334 °N, 105.841 °W, 2796 m above
sea level) is a small, deep lake (c. 1.5 ha surface area; 7.9 m depth) in a lodgepole pine-
dominated subalpine forest in Rocky Mountain National Park. The even-aged forest surrounding
the lake dates to a 1782 CE fire (Sibold et al., 2007). Mean monthly temperature is -8.5 °C in
January and 14 °C in July, and average total annual precipitation is 483 mm (Western Regional
Climate Center 1940-2013 observations, from Grand Lake, CO). Detailed methods for the
collection and analysis of this record are found in Dunnette (2014). Briefly, the 4500-year record
has an average sample resolution of four years, and a chronology constrained by 25 accelerator
mass spectrometry $^{14}$C dates and 13 $^{210}$Pb dates spanning the upper 20 cm. Pollen analysis
indicates that the site was continuously dominated by lodgepole pine for the duration of the
record presented here, with successional changes following inferred fire events. Dunnette (2014)
used macroscopic charcoal and magnetic susceptibility (a soil-erosion proxy) from Chickaree
Lake to infer the timing and severity of wildfires, identifying "high-severity catchment fires"
(those with associated erosion) and "lower severity/extralocal fires" (those without associated
soil erosion). Thus, while all fire events were likely stand-replacing, the difference between these
two fire types was the association with soil erosion. Here, we use the Chickaree Lake fire history
record to inform the disturbance component of the DayCent ecosystem model by prescribing the
timing and severity of past fire events within a simulated lodgepole pine-dominated subalpine
forest.

### 2.2 Model description

DayCent is the globally recognized daily timestep version of the biogeochemical model
CENTURY, widely used to simulate the effects of climate and disturbance on ecosystem
processes including forests worldwide (Bai and Houlton, 2009; Hartman et al., 2007; Savage et



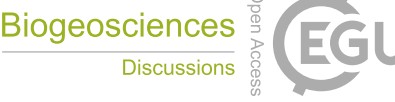

al., 2013). DayCent is a logical choice for our purposes, because it includes soil C pools that
have long turnover times, spanning months to 4000 years, and thus can represent long-term
ecosystem change.
Required inputs for the model include vegetation cover, daily precipitation and temperature, soil
texture, and disturbance histories. DayCent calculates potential plant growth as a function of
water, light, and soil temperature, and limits actual plant growth based on soil nutrient
availability. The model includes three soil organic matter (SOM) pools (active, slow, and
passive) with different decomposition rates, above and belowground litter pools, and a surface
microbial pool associated with the decomposing surface litter. Plant material is split into
structural and metabolic material as a function of the lignin to nitrogen ratio of the litter (more
structural with higher L:N ratios). The active pool (microbial) has short turnover times (1-3
months) and the slow SOM pool (more resistant structural plant material) has turnover times
ranging from 10 to 50 years depending on the climate.  The passive pool includes physically and
chemically stabilized SOM with turnover times ranging from 400 to 4000 years. For this study,
DayCent was parameterized to model soil organic carbon dynamics to a depth of 30 cm.
Disturbances in DayCent are prescribed and can be parameterized to reflect "severity" through
associated impacts to the ecosystem (e.g., biomass killed, nitrogen lost, soil eroded). Model
outputs include soil C and N stocks, live and dead biomass, above- and below-ground net
primary productivity (NPP), heterotrophic and autotrophic respiration, fire emissions, and net
ecosystem production (NEP, defined as the difference between NPP and heterotrophic
respiration). We define net ecosystem carbon balance (NECB) as the difference between NEP
and fire emissions.
**2.3 Model parameterization**
DayCent submodels associated with tree physiological parameters, site characteristics, soil
parameters, and disturbance events were modified using available site-specific observations
(Dunnette et al., 2014; Sibold et al., 2007), values from the literature (Kashian et al., 2013;
Turner et al., 2004), and publically available climate and soils databases. Climate data required
for DayCent include daily minimum and maximum temperature and precipitation which were
obtained for a 30-yr period from DAYMET (Thornton, 2012). For all model runs, the 30-yr





climate dataset was "recycled" for the duration of the run; thus, unless specified by a scenario
name, climate was functionally non-varying over the duration of the simulations (beyond the
variability in the 30-yr dataset). Soil texture and classification were identified using the NRCS
SSURGO database (NRCS, 2010).  Model input and parameterization files are available for
download as supporting information files.
We defined two types of stand-replacing fire to distinguish between the two types of fires
identified in the paleo record. High-severity catchment fires from the paleo record were
simulated by 95% tree mortality and a soil erosion event with ~1 Mg ha$^{-1}$ of soil loss from the
watershed (Miller et al., 2011); we refer to these as high-severity fires with erosion. Lower-
severity/extra local fires from the paleo record were simulated by 95% tree mortality with no
associated soil-erosion event; we refer to these as high-severity fires without erosion. Thus, the
key difference between the two fire types simulated is the associated soil erosion. After
parameterization, we evaluated modeled aboveground NPP, soil C, total ecosystem carbon, and
disturbance C losses against observations of similar-aged lodgepole pine stands in the Central
Rockies ecoregion (Hansen et al., 2015; Kashian et al., 2013; Turner et al., 2004).
**2.4 Model experiments**
We performed a series of modeling experiments to address our questions using the Chickaree
Lake paleo-fire record, varied disturbance histories, and varied climate (Table 1). First, DayCent
was 'spun up' and equilibrated to soil C and NPP levels characteristic of mature lodgepole pine
stands in the region with a constant return interval of 145 years between high-severity fires with
erosion, replicating the estimated fire rotation period (and mean fire-return interval) for the
broader study area (Sibold et al., 2007). This spinup period lasted for 2000 years, and it
represents what would be done for model use, in the absence of the long-term fire history
information from the paleo record. All experimental simulations were extended from this spinup
equilibrium simulation starting 4500 years before present (BP, where "present" is 1950 CE) and
running through 2010 CE, for a total of 4561 simulation years. We defined our model simulation
that would normally be used in the absence of paleo-informed disturbance histories ("equilibrium
scenario") as a continuation of the equilibrated spinup with the same climate and fire regime,
with only the last known fire event (1782 CE) explicitly simulated.



In addition to this equilibrium scenario, we implemented four additional scenarios that together
helped illustrate the duration, magnitude, and relative importance of fire-induced changes to
forest biogeochemistry. (1) To test the impacts of variability in fire timing and severity on
important biogeochemical states, we compared the equilibrium scenario to a "paleo-informed
scenario," which had a mean fire return interval of 120 years for all fires, and 334 years for the
high-severity fires with erosion. (2) To identify the duration of a legacy effect from fire-regime
variability, we constructed eight "partially paleo-informed scenarios", which included
increasingly longer periods of information from the paleo-fire record, spanning the past 500 to
4000 years, in 500-year increments that ended in 2010 CE ("$Paleo_{500}$", "$Paleo_{1000}$", …,
"$Paleo_{4000}$"; Figure 1a). (3) To identify how a systematic shift in fire frequency would impact
carbon balance, we created two additional scenarios with shortened and lengthened fire return
intervals. Beginning with the observed paleo-fire record, we modified each interval between fires
to be (a) shortened by 25% ("Increased fire frequency") or (b) lengthened ("Decreased fire
frequency") by 25% (Figure 1b). The corresponding mean fire return intervals of these two
additional runs were (a) 90 years and (b) 155 years. (4) Finally, to place the impacts of fire-
regime variability into the context of projected future climate change, we compare results to both
paleo-informed scenarios and equilibrium scenarios that included a constant 2 °C increase in
temperature (Figure 2; "Equilibrium + 2 deg C"). Specifically, we increased the minimum and
maximum daily temperatures of the DAYMET climate record for May through September by 2
°C, representing a very simple growing-season warming scenario. Because the fire events are
decoupled from climate, the prescribed warming did not impact the fire history. While we
recognize that fire and climate are closely coupled, these scenarios are considered experiments
that reveal the impacts of warming alone. The relative difference between the two scenarios (e,
paleo-informed and equilibrium with warming) and the equilibrium scenario is used to gauge the
relative impacts of fire-regime variability vs. warming on carbon balance.
We evaluated the results from each scenario in terms of modern end points of soil C, soil N, and
NECB as well as total cumulative changes in NECB over the entire record. We define
cumulative NECB as a running total, such that the sum at any given year represents the
integrated impacts of past disturbance events. For example, when return intervals between
disturbance events are shorter than C recovery times, cumulative NECB will remain negative.
Finally, we considered uncertainty in our estimates based on the uncertainty in the reconstructed



fire history record and our assumptions about soil erosion. While there is also uncertainty
associated with modeled estimates of soil C, NECB, and other C fluxes presented, we are not
attempting to provide estimates that are any more precise than measured modern states (e.g.
STATSGO derived soil C). Rather, we compare the variability in ecosystem states arising from
fire-regime variability to the uncertainties in the model that are revealed when evaluated against
modern observations from the literature.

## 3    Results and Discussion

### 3.1 Model parameterization and evaluation

Our modeled estimates of modern soil C (to 30 cm) of 54 and 62 Mg C ha$^{-1}$, for the
equilibrium and paleo-informed scenario, respectively (Figure 2), compare well with STATSGO
(NRCS, 2010) estimates of 66 ± 16 Mg C ha$^{-1}$ for the Chickaree Lake region, and with
measurements of current soil C (to 30 cm) ranging 51 to 73 Mg C ha$^{-1}$ in similarly aged (> 200
year) Rocky Mountain *Pinus* stands (Bradford et al., 2008). Modeled estimates of aboveground
NPP were also in agreement with observations averaging 156 and 172 g C m$^{-2}$ for the
equilibrium and paleo-informed simulations, respectively, compared to estimates from the
Northern or Central Rockies ranging from 100 to 200 g C m$^{-2}$ (Hansen et al., 2015). Finally, fire
emissions from our modeled estimates range from 20 to 30% loss of aboveground C, broadly in
agreement with other studies (Campbell et al., 2007; Smithwick et al., 2009).

### 3.2 Fire-regime variability impacts soil C and NECB

When DayCent was driven with the paleo-informed fire history, soil C accumulation was
8 Mg ha$^{-1}$ more at the end of the simulation than in the equilibrium scenario (Figure 1). Total
NEP summed over the 4561-year period was also higher in the paleo-informed scenario (1276
Mg C ha$^{-1}$) compared with the equilibrium scenario (1171 Mg C ha$^{-1}$), directly reflecting NPP
rates that were higher than heterotrophic respiration (Figure 3, black bar). In the paleo-informed
scenario, cumulative emissions due to combustion losses (i.e., "fire emissions") were lower than
NEP over the entire record, resulting in a cumulative NECB of 27 Mg C ha$^{-1}$ more than the
equilibrium scenario (Figure 3; black bars).
The paleo-informed scenario showed substantial variability in soil C (Figure 2) and
NECB (Figure 4) trajectories, and higher total accumulations relative to the equilibrium scenario.
In fact, the range of variability in soil C over the paleo-informed simulation, from c. 45 to 65 Mg



C ha$^{-1}$, nearly spanned the range of observations of current soil C (to 30 cm) in similarly aged (>
200 year) Rocky Mountain *Pinus* stands (Bradford et al., 2008). For the first ~2000 years of the
paleo-informed scenario, long-term mean soil C was similar to baseline levels of soil C in the
equilibrium scenario (Figure 2), averaging around 54 Mg C ha$^{-1}$, though with substantial
variability on centennial time scales. Following this period, the soil C trajectory increased
distinctly in the paleo-informed scenario during a 500-year period with only one high-severity
fire without erosion (c. 2500 cal yr BP). Despite a return to a mean fire return interval closer to
the equilibrium scenario, soil C persisted at this elevated level for the following 2000 years (c.
2000 cal yr BP to present), resulting in 8 Mg C ha$^{-1}$ (15%) more than the equilibrium scenario at
the end of the simulation (2010 CE). A similar trend was observed for NECB (Figure 4), where
the paleo-informed scenario maintained a lower NECB in the first half of the record compared
the second half. In the latter half of the record, NECB was more consistently positive, ultimately
storing more ecosystem C than the equilibrium scenario. The dynamism in NECB over time is
consistent with the findings of Kelly (2016). Together, this work highlights the value of
examining the ecosystem impacts of past fire-regime variability, which may include disturbance-
free or intensified disturbance periods that are not currently represented in or predicted by
ecosystem models.

### 3.3 Impacts of fire-regime variability last for millennia and can outweigh climate impacts

We compared the partially paleo-informed scenarios to the equilibrium scenario to determine the
length of time necessary to arrive at the same inferences about soil C and NECB (i.e., endpoints
as totals) as in the full paleo-informed scenario. We found that disturbance-regime legacies
lasted for millennia. The number of years needed to simulate the CE 2010 values was between
2000 and 2500 years (Figure 5). Specifically, total NECB and soil C (endpoints that serve as
initial conditions for future modeled states) were nearly the same when using 2500 to 4500 years
of the paleo-fire record, but differed by more than 1 Mg C ha$^{-1}$ when using only 500 to 2000
years of the paleo-fire record.
Differences between the paleo-informed and equilibrium scenario were an order of magnitude
greater than differences between the equilibrium scenarios with and without a uniform 2 °C
warming during the growing season. Warming resulted in a small net decrease in soil C of 0.3
Mg C ha$^{-1}$, and a reduction in NECB by 0.2 Mg C ha$^{-1}$ relative to equilibrium scenario. Warming




with a constant fire-return interval resulted in a small proportional increases in both NPP and $R_h$,
while NEP did not change.
Our results imply that C dynamics in lodgepole pine forests are far more sensitive to variability
in the timing and severity of fire activity than to changes in climate. This inference is also
consistent with findings from strand-replacing fire regimes in Alaskan boreal forests, where C
dynamics over the past 1200 years were more strongly shaped by fire activity than by climate
variability (Kelly et al., 2016).
**3.4 Implications for projecting future ecosystem states**
We varied the paleo-informed disturbance regimes by increasing and decreasing the frequency of
events by 25% to evaluate the effects of changing fire regimes. As expected, increased fire
frequency (i.e., shorter return intervals) resulted in a cumulative loss of ecosystem C compared
to equilibrium and paleo-informed scenarios, with NECB 13 Mg C ha$^{-1}$ lower over the entire
simulation period (Figure 3), and with periods of net carbon loss lasting nearly 800 years (Figure
4; red line). The losses reflect large increases in fire emissions, without concurrent proportional
increases in NEP (Figure 3). In contrast, with decreased fire frequency (i.e., longer return
intervals), NECB increases by 67 Mg C ha$^{-1}$ compared to equilibrium, and by 40 Mg C ha$^{-1}$
compared to the original paleo-informed scenario. Again, this is primarily due to an unbalanced
increase in NEP compared to fire emissions (Figure 3).
While the differences in NECB (27 Mg C more) and soil C (8 Mg C more) between the paleo-
informed and equilibrium scenarios are ultimately small for this single watershed, the impact of
fire-regime variability will depend on the synchrony of events at the regional and sub-continental
scales (Kelly et al., 2016). This is especially important when considering the trajectory of NECB
compared to equilibrium simulations during the periods of the paleo record when fire frequency
or severity were higher than in the past few centuries. Cumulative NECB was negative, serving
as a net source of C to the atmosphere, for periods of up to 500 years in the paleo-informed
scenario and up to 1000 years under scenarios with increased fire frequencies.
Given the strong correspondence between observed and simulated modern C stocks, uncertainty
in our estimates of past carbon dynamics stems primarily from uncertainty in the timing and
severity of past fires. The fire history reconstruction has an estimated temporal precision of





several decades (±10-20 years) (Dunnette et al., 2014), but because C dynamics unfold over

centuries to millennia, this level of uncertainty has negligible effects on our inferences. The more

important source of uncertainty is the potential for false positives or false negatives in the fire

history reconstruction: failing to detect a fire that occurred in the past, or identifying a fire that

did not affect the Chickaree Lake watershed. While the Chickaree Lake record clearly identified

the most recent high-severity fire in the watershed (Dunnette et al., 2014), we cannot quantify

accuracy over the past four millennia. However, the range of variability in individual fire return

intervals reconstructed at Chickaree Lake (20-330 year) is consistent with the range of intervals

reconstructed from other lake-sediment records in Colorado subalpine forests (Calder et al.,

2015); 75-885, 45-750, 30-645, 30-1035 yr, (Higuera et al., 2014), suggesting that the C

dynamics highlighted here are not unique to this single fire history reconstruction.

In addition to fire timing, simulated C dynamics were also a function of variability in fire

severity, which in this study reflects the degree of soil erosion associated with stand-replacing

fire events. Watershed soil C losses were partially driven by the erosion events accompanying

the "high severity catchment fires" reconstructed in the paleo record. Because we have

prescribed both fire and erosion, we cannot predict the range of soil C loss that may occur due to

changes in precipitation regimes or if any erosion occurs with the lower severity events;

however, these results provide an estimate of expected changes in soil C for at least the higher

severity events. With expected changes to future precipitation regimes, including intensification

of rain events that could lead to increased erosion following fire (Larsen and MacDonald, 2007;

Miller et al., 2011), ecosystem model development should include prognostic erosion to account

for variability in this ecosystem process, especially at regional scales.

Finally, while simulated past carbon dynamics are also limited by the lack of paleoclimate data

driving DayCent, our results suggest that C dynamics are much more sensitive to the timing and

severity of fire events than to even relatively large changes in climate (e.g., 2 °C warming).

Further, because we have decoupled climate from fire by using prescribed fire events, the lack of

a paleoclimate does not affect our conclusions about the impacts of fire-regime variability on C

balance. While we used the paleo-informed modeling scenarios to test general hypotheses about

the impacts of fire-regime variability on biogeochemical dynamics, future efforts to simulate the



coupled climate-fire-ecosystem dynamics of the past clearly require independent paleoclimate drivers.

## 4 Summary and Conclusions

Our simulations highlight fire-regime variability as a dominant driver of C dynamics in lodgepole pine forests, with periods of unusually high or low fire activity creating legacies lasting for centuries to millennia. Anticipating the impacts of future climate or disturbance-regime change on forest carbon balance, therefore, should be done in the context of past variability, with the duration dependent on the frequency and variability of relevant disturbance processes. In the case of stand-replacing wildfires this requires information spanning at least several centuries, and at Chickaree Lake this required several millennia, well beyond the length of both observational and tree-ring records. While a number of studies have reported ecosystem impacts or recovery times from individual fire events and then extrapolated to infer scenarios that would lead to C gain or loss (Dunnette et al., 2014; Kashian et al., 2013; Mack et al., 2011; Smithwick et al., 2009), our paleo-informed scenario highlights the importance of variability in fire timing and severity for carbon cycling dynamics, independent of complete shifts in a fire regime.

Our findings also have implications for Ecosystem and Earth system model development, which are increasingly including prognostic fire components (Lasslop et al., 2014), primarily driven by climate and fuels. Some models are also representing post-fire C and N dynamics beyond simple combustion of woody pools. Development of these modules depends on observations of fire and climate interactions, fuel availability, and post-fire C and N dynamics. We suggest that this requires accurately accounting for the (often high) variability inherent in stand-replacing fire regimes, independent from or in response to climate variability. Our results indicate that even utilizing tree-ring record that span several centuries may not be sufficient to capture this variability. Further development of prognostic (predictive) fire processes in ecosystem models would benefit from the use of paleo-fire records to evaluate fire occurrence and severity, and if combined with paleoclimate data, model algorithms could be further improved to accurately reflect past variability.



The importance of fire-regime variability in determining ecosystem C dynamics implies that equilibrium scenarios are a poor assumption for conceptualizing and simulating fire regimes in ecosystem and Earth system models. Particularly at spatial scales larger than an individual site, such a simplification may result in C-balance projections that are grossly overestimated or underestimated. We demonstrate how variability in the timing and severity of disturbances can potentially have long-lasting and compounding impacts on ecosystem states, such that modern (or future) states can reflect dynamics that have unfolded over centuries to millennia. For our modeling scenarios in lodgepole-pine dominated forests, the effects lasted approximately 2500 years. The duration of these legacies will depend on the ecosystem, and the degree of variability in disturbance frequency and severity, relative to an equilibrium scenario. Ultimately, the implications of fire-regime variability on biogeochemical states will depend strongly on the synchrony of fire activity across spatial scales larger than a single watershed. If fire activity is synchronized at landscape to regional scales, as in past (Calder et al., 2015; Marlon et al., 2012; Morgan et al., 2008) and as anticipated for the future (Westerling et al., 2011) in the Rocky Mountain forests, we would expect to see similar centennial- to millennial-scale dynamics in biogeochemical states revealed here, which would have important implications for carbon cycling, including potential feedbacks to $CO_2$ induced warming.

## 5 Data Availability

The following datasets are available at Dryad.org <url TBD>: the fire history record generated from the charcoal record, the relevant model output, and model input files and climate input file.

*Author Contributions.* T.W. Hudiburg and P.E. Higuera designed the study, analyzed the data, and prepared the manuscript with contributions from J.A. Hicke.

*Competing interests*. The authors declare that they have no conflict of interest.

*Acknowledgments.* We thank K. McLauchlan and B. Shuman for valuable discussions on these topics. T.H. was supported by the NSF Idaho EPSCoR Program and by the National Science Foundation under award number IIA-1301792. P.E.H was supported by the National Science Foundation under award number IIA-0966472 and EF-1241846, and JAH was supported by the Agriculture and Food Research Initiative of the USDA National Institute of Food and


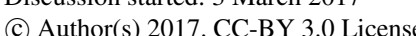


Agriculture (Grant 2013-67003-20652) and the National Science Foundation under award
number DMS-1520873. The authors declare no competing financial conflicts of interests or other
affiliations with conflicts of interest with respect to the results of the paper.

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



## Tables

**Table 1.** Model simulation scenarios, including climate, fire regime, duration, and summary description.

| Scenario | Purpose | Climate* | Fire Regime | Duration (yr) | Description |
|---|---|---|---|---|---|
| **Spinup** | Spin up C, N pools to equilibrium conditions | Ambient | Fixed 145-yr return interval; high severity with erosion | 2000 | DayCent initialization run for NPP and C to reach equilibrium conditions. |
| **Equilibrium** | Run with fixed fire interval | Ambient | Fixed 145-yr return interval; high severity with erosion | 4561 | Equilibrium run extended from the spinup run for the length of the paleo-fire record. |
| **Paleo-Informed** | Run with observed paleo-fire intervals and severity | Ambient | Paleo-record; high severity with and without erosion | 4561 | A 4561-year simulation with fires matching the timing and severity from the paleo-fire record. |
| **Increased fire frequency** | Run with paleo-fire intervals decreased by 25% | Ambient | Modified Paleo-record; 90-yr MFRI with high severity with and without erosion | 4561 | A 4561-year simulation with the timing between fires in the paleo-informed scenario decreased by 25%. |
| **Decreased fire frequency** | Run with paleo-fire intervals increased by 25% | Ambient | Modified Paleo-record ;155-yr MFRI with high severity with and without erosion | 4561 | A 4561-year simulation with the timing between fires in the paleo-informed scenario increased by 25%. |
| **Paleo$_{500}$… Paleo$_{4000}$** | Test influence of length of paleo record on modern states | Ambient | Paleo-record; high severity with and without erosion | 500 - 4000 | Branches from the equilibrium scenario at varying points in time, in 500-yr increments**. All scenarios ends in CE 2010. |
| **Spinup_ 2deg** | Same as Spinup but under warming scenario | + 2 ℃ | Fixed 145-yr return interval; high severity with erosion | 2000 | DayCent initialization run for NPP and C to reach equilibrium conditions, with uniform warming. |
| **Equilibrium_ 2deg** | Same as Equilibrium but under warming scenario | + 2 ℃ | Fixed 145-yr return interval; high severity with erosion | 4561 | Equilibrium run extended from the spinup run for the length of the paleo-fire record, with warming. |
| **Paleo-Informed_ 2deg** | Same as Paleo-Informed but under warming scenario | + 2 ℃ | Paleo-record; high severity with and without erosion | 4561 | 4561-year simulation extended from the spinup run, with fires matching the timing and severity from the paleo-fire record, with warming. |

\* 30-year recycled historical record (DayMet)

\*\* For example, the 500 year simulation starts in the year 1510 (CE) and runs until the end of 2009





**Figures**

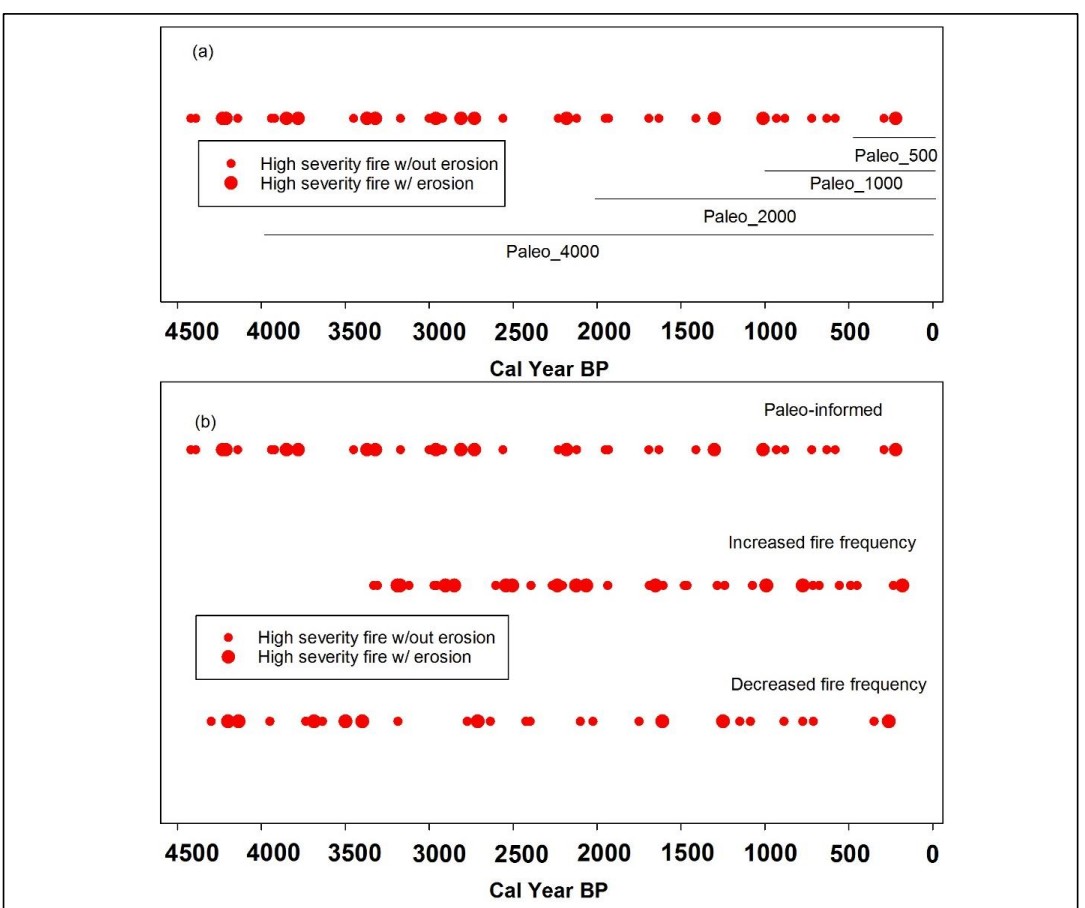

**Figure 1.** Paleo-informed fire history scenarios used to drive the DayCent model. (a) Fire history
record form Chickaree Lake (red dots), with horizontal lines illustrating the duration of the
record used in the incremental "partial paleo-informed" scenarios (Paleo_500…4000). (b) The
same full Chickaree Lake fire history record used in the paleo-informed scenario (top), with the
two additional scenarios representing a 25% increase and 25% decrease in fire frequency
(bottom two scnearios).




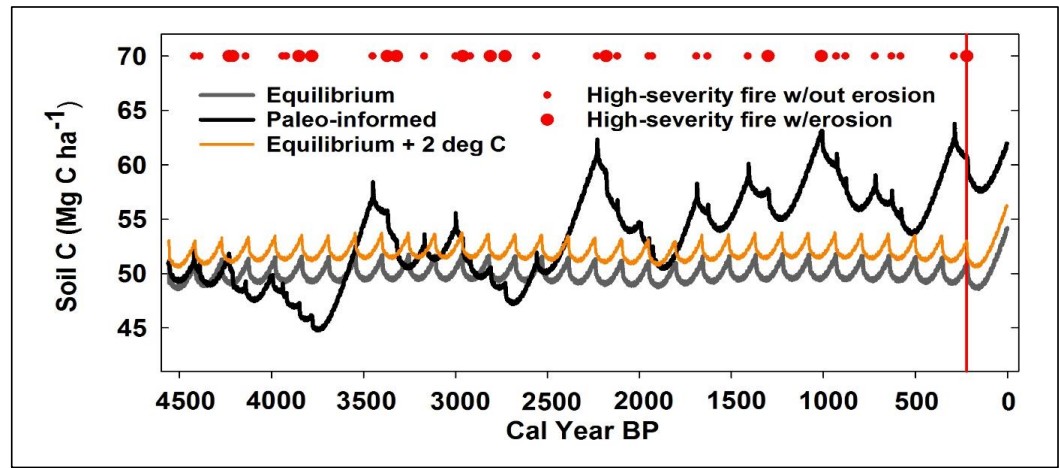

**Figure 2.** Model simulations of equilibrium (grey), equilibrium plus a 2 ℃ warming (orange),

and paleo-informed (black) total soil carbon (C) in Mg C ha$^{-1}$. Each simulation branches from a

2000-year equilibrium spinup starting at the same soil C baseline and runs for 4561 years (4500

BP to CE 2010). Values for the warming scenario were increased by 2 Mg C ha$^{-1}$ to be

distinguishable from the equilibrium scenario. The large red dots represent the years of the high-

severity fires with erosion, and the small red dots are high-severity fires without erosion used to

drive the paleo-informed model run. A constant 145-year fire return interval was used for the

equilibrium run. The vertical red line indicates the most recent stand-replacing fire (1782 CE),

reconstructed from the tree-ring record (Sibold et al., 2007).





534

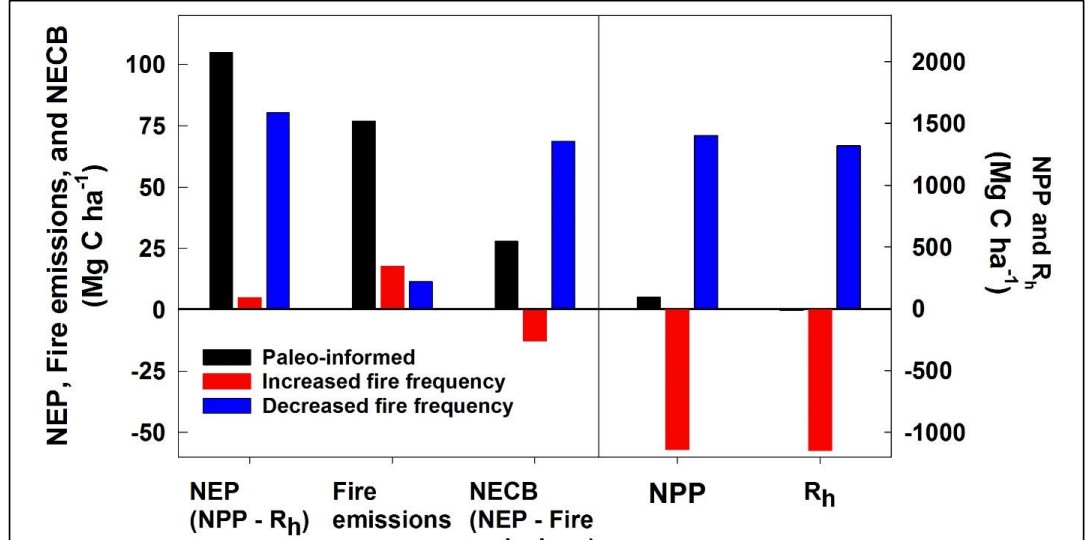

535

**Figure 3.** Accumulated anomalies in fluxes relative to equilibrium scenario, in Mg C ha$^{-1}$,

summed over the entire 4561-year simulation period. NEP, fire emissions, and NECB (left y-

axis) and NPP and Rh (right y-axis) for the paleo-informed (black), increased fire frequency

(red; 155 year mean FRI), and decreased fire frequency (blue; 90 year mean FRI) scenarios.

Negative (positive) numbers indicate a decrease (increase) in total carbon flux compared to the

equilibrium scenario.
















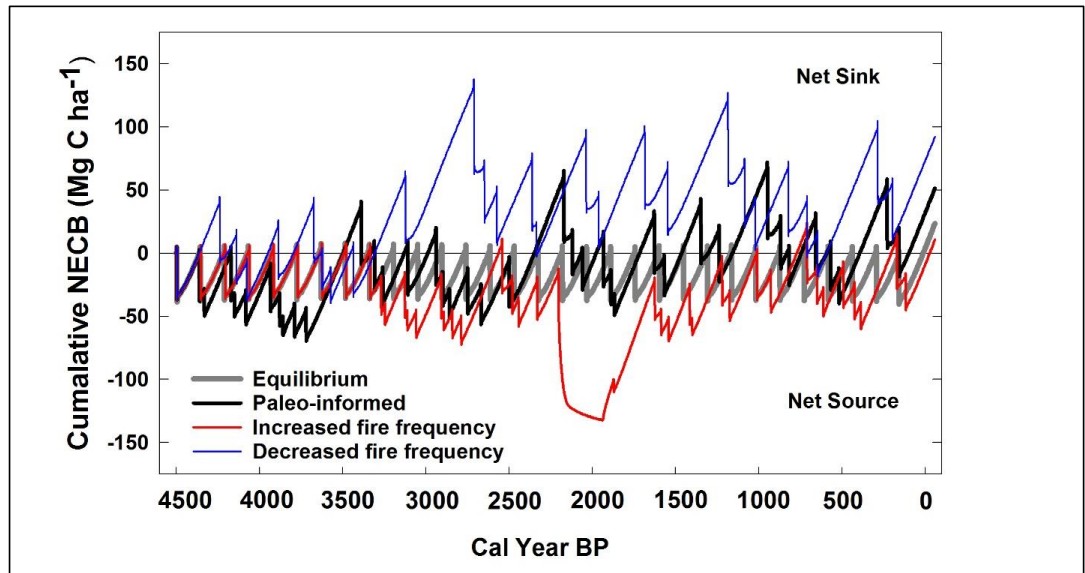


**Figure 4.** Trends in cumulative net ecosystem carbon balance (NECB) over time for the

equilibrium, paleo-informed, increased fire frequency, and decreased fire frequency scenarios

over the last 4561 years. Positive numbers indicate a cumulative net sink while negative

numbers indicate a cumulative net source.

559





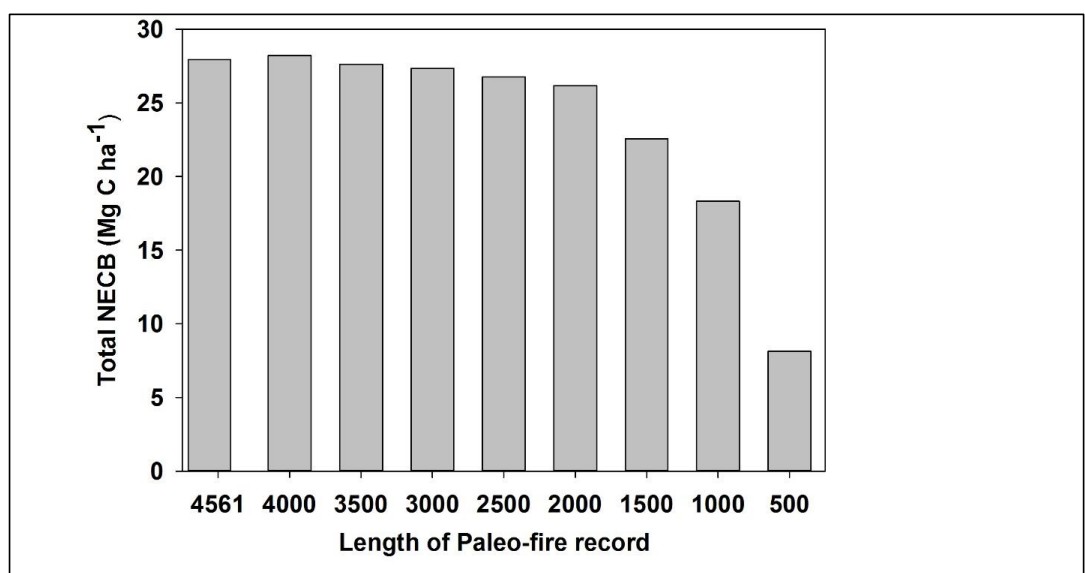

**Fiure 5.** Total NECB (NPP - Rh - fire emissions) for the 4561-year simulated period and for
each of the partially paleo-informed scenarios (Paleo_500, Paleo_1000, etc. in Figure 1). Each
partially paleo-informed scenario branches from the equilibrium scenario in the year indicated on
the x-axis. For example, the 500-year record only includes fires that occurred in the most recent
500 years of the paleo-fire record (1511-2009 CE).