# Peer review of "Fire-regime variability impacts forest carbon dynamics for centuries to millennia"

_Biogeosciences, 2017_

## Short Comment (SC1) · 29 Mar 2017

*A note upfront from the submitting person: This review was prepared by two master students (Lena Weiss and Fatemeh Ajallooeian) in geography or earth system science at the University of Zurich. The review was part of an exercise during a second semester master level seminar on "the biogeochemistry of plant-soil systems in a changing world", which I organize. We would like to highlight that the depth of scientific knowledge and technical understanding of these reviewers represents that of master students. We have enjoyed discussing the manuscript in the seminar, and hope that our comments will be helpful for the authors.*

In this paper, the authors investigate how fire-regime variability impacts soil C and net ecosystem carbon balance. These events are of importance in biogeochemical

processes including net carbon (C) balance. In long-term studies and monitoring of forests, biogeochemical cycles represent a great resilience against long lasting stand-replacing fire events. On the contrary, the aftermath of such repeated fires or temporal fluctuations in a fire regime (e.g., the characteristic timing or severity of fire) are not known very well. In theory, such fire events and variabilities can influence carbon balance for centuries to millennia. The authors found that C range in a paleo-informed scenario is significantly different from an equilibrium scenario (with a constant fire return interval), likely because of the difference in fire severity and timing variabilities. Paleo-informed scenarios consisted of multi-century periods of positive and negative net ecosystem C balance, with the amount of net C balance being largely greater than the equilibrium scenario. In addition, these fluctuations produced long lasting effects on the C balance for millennia. This implies that fire-regime variability is a major driver of C trajectories in stand-replacing fire regimes. Thus, anticipating carbon stability in these systems, will depend strongly on the ability of ecosystem models to represent a realistic range of fire-regime variability over the past several centuries to millennia.

Overall the paper is well structured and easy to follow. The importance of the theme is highlighted in several occasions. Methods, material, applications are well explained. Some things might need a more accurate definition for a better understanding. Using a three-pool model makes it more valuable and leads to more accurate results. Main positive point is the declaration of uncertainties and suggestions for future projects.

Two minor errors in grammar were found:

- Page 2, line 48: we suggest changing the word "great" to "greater" since it is followed by the word "than" and in comparison, certain adjectives such as great should get an "er" or "est" at the end.

- Page 4, line 83: we would change "significance influence of fire" to significant influence since it makes more sense

For a better understanding and conception, we suggest the following: - Page 2, line 40:

we would find a definition of "C trajectories" helpful

- Page 3, line 61: it is somewhat unclear what the authors mean by pool sizes, we suggest that they indicate which elements pool sizes they specifically mean (e.g. carbon or nitrogen or etc.,)

- Page 3, line 71: it is not clear what is meant by Net Ecosystem Carbon Balance (NECB)

- Page 4, line 86: the term "spin up" is confusing. We suggest that the authors try to explain and clarify this term in a more understandable wording perhaps by defining this term with a simple example before using it.

- Page 5, line 139-141: "Day Cent" Is well described but already mentioned in section 2.1, therefore we suggest the description should come earlier

- Page 6, line 151-152: is L:N and lignin to nitrogen the same? It is not mentioned in the text

- Page 7, line 182: from our point of view, the "key difference" between the two fire types should come at the beginning of the paragraph

- Page 8, line 208: timeframe CE, is that defined as common era?

- Page 8, lines 211-219: we think the explanation of different scenarios can be expressed in a more precise and separated way. The description of additional scenarios make it difficult to understand and follow the subject since they're told altogether. Perhaps by separating the scenarios and explaining each of them on an independent paragraph, the concept can be easier to follow. The use of that many brackets makes it more confusing than helping anything.

- Page 9, line 248: isn't the data fitted? Not surprising that it is "broadly in agreement"

- Page 13, line 360-365: very long and complicated sentence. We would suggest making more than one sentence out of it for a better understanding

- Page 13, line 369: the word "woody pool" should be clarified

- Page 14, line 383 & 388: are "ecosystem states" and "biogeochemical states" the same? Here we would need simplification or a better definition

Concerning the figures: - Implement results in Table 1

- Figure 1: For a better visual understanding, it would be nice to have at least two different colors for the different types of fire. Also, different symbols could be used. The spacing between the line is very big and could be better used. It would be sufficient to have only one legend as it is the same, and we can read the word "high severity fire" four times in a small figure. That could be simplified.

- Figure 2: It is too confusing that the grey Equilibrium line and the yellow Equilibrium + 2 degrees have the same value on the y-axis but it's not shown.

- Figure 1, 2 and 4: In the text the time data is in CE. In the Figures time data Cal BP is used. We would suggest to only use one time specification.

---

## Referee Comment (RC1) · Anonymous Referee #1 · 20 Apr 2017

Review of Biogeosciences manuscript #

As my review comes after the publication of M.W.I. Schmidt's comment, I used part of it when I considered that students who performed the reviewing training made constructive comments and added my own ones elsewhere.

This study is overall a good example on how prescribed fire regime, reconstructed from paleoecological data, can be used as input into a biogeochemical model in order to answer to several questions related to the long term dynamics of ecosystems. Authors wanted to see 1) whether past changes in fire variability impact biogeochemical processes and therefore soil C and NECB as compared to information from only one fire event (last one) or several ones but under a regular assumed fire regime (fixed), 2) for how long such impacts could stand, and 3) to assess the relative roles of fire and climate changes on such impacts on C stocks and fluxes.

Such modelling experiments are necessary to understand processes at play and to likely disentangle the impact of the different factors. The well-documented site used in the present study is a valuable data source resulting from the combined results of several studies that targeted different proxies over time with associated expertise. Therefore, such approach, while still rather rare, should be encouraged. Despite these general positive comments, there are few issues that need to be clarified, and that, in turn will require to remove the third objective or at least to present its results differently.

First of all, even though the authors refer to past published studies, they should present or document the reconstructed response of vegetation (changes or not) the site recorded at least with the same level of information as for the fire reconstruction they provide.

Secondly, and most importantly, I wonder why authors have used only the same fixed 30-year time series for climate data whatever the time frame simulated over the last 4500 years BP instead of using past climate simulations from GCM or ESM whose many have Holocene climate as well as Future climate runs. This would have prevent authors from saying that fires and climate are disconnected which is absolutely not true, or at least need to be tested for each ecosystem studied. Moreover, instead of just increasing the 30-year time series temperature by 2°C, they could have used the full climate time series for the 21st century simulated by the same climate or earth models that provided the Holocene runs. They even could have tested different IPCC scenarios and their impact of the NECB. The use of climate model data would have provided precipitation time series as well, whose changes could also have impacted soil nutrient (and C) leaching. Indeed, it is easy to show that fire regime change outweighs climate change when such climate change may be unrealistic or only taken into account through temperature increase whereas several studies have documented and discussed about the potential counter-effect of precipitation increase in compensating the effect of temperature increase on fire occurrences and spread. It is even more important in the studied system as authors suggested and used two types of high severity fires: those with and those without erosion. Stand-replacing fires (95% mortality) are not really severe fire if post-fire regeneration is occurring in the next following years from naturally adapted species. Fire severity would rather refer to the difficulty of post-regeneration encountered in special cases. Stand-replacing fires are usually very intense and fuel consumption includes all the litter and humus layers, leaving the mineral soil exposed. So, if erosion in the burned watershed occurs (towards the lacustrine receptacle), it is performed during (heavy) rainfall events. Therefore, this is another argument to show that it would have been valuable to use past simulated precipitation over the last 4500 years BP, in order to test if rainfall (even as mean annual rainfall) changes could have occurred contemporaneously to erosive events just after some fires as compared to others. Moreover, authors provide no information on the vegetation compartment modeled except the Net Ecosystem

Production for outputs, so we have no idea about which plant types are used for this site nor why 30cm deep was chosen as the targeted depth to analyze the site response. Finally, in the current version, except from NEP, we have not idea about the effect of vegetation change in terms of composition nor structure through time, we cannot see the direct as well as indirect effects of climate change on vegetation nor climate on fire as climate dataset was fixed and repeated along the 4500 years BP, even though fire ignition and fire spread conditions may have been more or less favorable.

For all these reasons I see two options that require to modify the manuscript:

Option 1: to do the modelling experiment exercise once again but using climate data that represent the studied Holocene period for the first part and the 21st century for the second part. Even though climate data come from GCM and are not perfect, they will still be better than present-day ones applied to past and/or future periods, especially if climate is tested and its relative impact compared to that of fire regime variability. In parallel to temperature and precipitation datasets, authors should explain how they deal with air $CO_2$ concentration as it should have been modified from 280 ppmv until 1750 to the historical recorded concentration until nowadays, and for the Future, at least a mean CO2 increase should be used if authors do not want to test several RCP scenarios. By keeping the CO2 at a fixed concentration could still be acceptable but once more, as they are tracking C pools, I think that the atmospheric C input should be taken into account.

Option 2: keep the modelling experiment in the current version but authors need at least to remove the third objective as climate has not been properly taken into account as compared to the fire regime factor. In such case, they should explicitly present this study as a first-step modelling approach integrating only the fire regime information and therefore only testing it. All sentences related to climate effect should be modified in order to rather present or discuss limit of non-using proper climate data. This would better fit with the balanced way results must be discussed. In such a case, the first two objectives are still OK. Results and conclusions should be fairly presented without omitting that the climate data used may be a limit to the interpretations done.

Otherwise, I found pertinent the improvements suggested in the M.W.I. Schmidt's comment posted for improvement definitions, more detailed explanations and improvement in figure quality so I encourage the authors to take them into account. They will facilitate the reading of the manuscript for people not fully familiar with model requirements and functioning such as the need of a spinup period, the use of several pools or compartments… If supplementary material is allowed I suggest to add such information there, even with a scheme presenting how the DayCent model works.

---

## Referee Comment (RC2) · Anonymous Referee #2 · 2 May 2017

The authors developed a comparative analysis in which paleofire data and an equilibrium (spin-up) approach are used to constrain the initial conditions of a biogeochemical model simulation. The logic for doing so is that the legacies of past fire disturbances are ignored when employing an equilibrium approach, thereby misleading the conclusion about time-dependent trajectories of NPP, soil C, etc, and strengths of carbon sinks and sources. This could be particularly true in a forest system where the disturbance regime is driven by pulses of extreme events, as opposed to one where disturbances occur at regular intervals. The authors illustrate this problem by providing an example from the Rocky Mountain National Park. This is an interesting topic and a fairly nice application of the problem. The paper generally reads well. Still, there are some aspects of the study that I'm more or less comfortable with. But I'm quite confident that authors can improve these aspects. Also, aside from discussing the biogeochemical elements,

it could be interesting to also compare some of the ecological attributes like age distribution of forest stands between the paleoinformed and equilibrium approaches. Clearly the distribution of ages will be quite different, which could have implications if eventually model simulations become a tool for forest management guidelines aiming at sustainability of ecological services.

Specific comments

Introduction:

L87-93 Would this rather illustrate that many models that perform a spin-up period lack a validation of their simulated biochemical cycle?

Materials and Methods:

L165 What exactly is the size of the simulated area? Are fires spatially-explicit? Or just based on random selection of cells? Perhaps a few word on this. L176 So climate and radiation are constant. This may be problematic because in the eventuality that climate was different during the late-Holocene, as compared to the Anthropocene, likely the simulation will be misleading the productivity levels. So I guess this is another argument for doing the +2C and -2C simulation experiments (L217-224). Not using paleoclimatic simulation is an important weakness of this study and I would recommend that authors put more emphasis on the importance of this temperature sensitivity analysis. However, they should note that temperature is not the only driver of NPP; radiation and precipitation are also important. L182-185 More details are needed in regard to the validation dataset. What kind of datasets are these observations? How were they derived? Why select these over others? What do you mean by 'similar-aged'?

Results and Discussion:

L241 What are the plus and minus signs for? Standard deviation or confidence intervals? What is the sample size? Area under analysis? Seems that crucial details are missing. L274-278 This statement about disturbance free or intensified disturbance

periods is partly false, because DGVMs now have the capacity to run fire dynamics using paleoclimate simulations that feed into a dynamic fire behaviour and growth model (e.g., LPJ-LMfire). This removes the necessity to do the paleo-informed, but nevertheless paleodata comparison is necessary as a validation step. L294-298 This is not really new and has been known for decades. The impact of fire versus vegetation is quite obvious considering that fire has the potential to exclude treed vegetation from landscapes despite generally improving growth conditions with warming and $CO_2$ L343 "the lack of paleoclimate data" : this is an important weakness of this study. A few sentences about this is needed here to help readers unfamiliar with this issue to understand what is meant by 'paleoclimate data'.

Figures:

Figure 4 This figure is not obvious to read. Perhaps put on separate panels.

---

## Referee Comment (RC3) · Anonymous Referee #3 · 10 May 2017

General comments

This paper examines patterns of forest ecosystem carbon dynamics in response to long-term past fire regime at the watershed scale. As noted by the authors, knowledge on this topic is scarce and the modeling exercises presented in the paper bring important new evidences that fire history left persistent legacies on ecosystem carbon trajectories on the centennial to millennial time scales, questioning the usual basal assumptions of ecosystem models. Globally, the text is clearly written, the scientific context and knowledge gaps are clearly exposed as the problematic and the general hypothesis. Also, the questions addressed here are very pertinent. That said, I advise the authors to follow previous comments and advises from SC1, RC1 and RC2. Moreover, a more deeper review of fire ecology with respect to carbon cycling could: i) help to better understand the choice of DayCent for this study; ii) bring a more critical inter-

pretation/discussion of the processes you mentioned (line 99-100) linked in DayCent model and improve the interpretation and discussion of the results. I also noted several improvement possibilities (see also Technical corrections): 1/ Strucure: Mixing results and discussion is sometimes confusing (especially for section 3.4). Because section 3.1 to 3.3 are not full discussions but rather descriptions and comparisons between your model estimates with values of other studies, it should not will be difficult to separate results and discussion. For example, discussion could contain a section on the limits, a section with the implications for projecting future ecosystem states and another for research development needs. 2/ Hypotheses: Based on Kelly et al. (2016), the general hypothesis assuming forest carbon budget modeling would be different between equilibrium runs and paleo-informed runs is explicit. Nevertheless, the alternative hypotheses that you mentioned (line 103) and results that were "expected" (line 301) are not explicitly described. You could add these hypotheses in the introduction. 3/ Model parametrization: According to SC1, DayCent is quite well described. Unfortunately, I was not able to access the model input and parametrization file. While is it clear that you informed the model with paleo-fire reconstruction from Dunette et al. (2014), it is less clear what you do with the vegetation data. You wrote that you "pair a paleoecological record of vegetation and wildfire activity" (line 98) and that DayCent requires input of vegetation cover (line 145), but no information is provided on vegetation in section 2.3. It would be important to get more details.

Specific comments

1. Does the paper address relevant scientific questions within the scope of BG? Yes. The paper deals with many fields within the scope of BG.

2. Does the paper present novel concepts, ideas, tools, or data? Yes. New data from modeling exercise based on previous works are presented.

3. Are substantial conclusions reached? Yes.

4. Are the scientific methods and assumptions valid and clearly outlined? Yes.

5. Are the results sufficient to support the interpretations and conclusions? Yes.

6. Is the description of experiments and calculations sufficiently complete and precise to allow their reproduction by fellow scientists (traceability of results)? Yes.

7. Do the authors give proper credit to related work and clearly indicate their own new/original contribution? Yes.

8. Does the title clearly reflect the contents of the paper? Yes.

9. Does the abstract provide a concise and complete summary? Yes.

10. Is the overall presentation well structured and clear? Yes, but could be improved (see General comments).

11. Is the language fluent and precise? Yes.

12. Are mathematical formulae, symbols, abbreviations, and units correctly defined and used? Yes, but see SC1 comments for [date] CE.

13. Should any parts of the paper (text, formulae, figures, tables) be clarified, reduced, combined, or eliminated? Yes. Values for equilibrium scenario should appear in Figure 3 or equilibrium scenario should be removed in lines 301-305. As the Chickaree Lake watershed is the object of this study, some characteristics such as the watershed size and topography (slope characteristics) could be mentioned. Moreover, you defined 8 partial paleo-informed scenarios but only 4 are represented in Figure 1. To facilitate the reading, I suggest to represent all partial paleo-informed scenarios in Figure 1 or you can specify that you show only 4 on the 8 scenarios in the figure caption.

14. Are the number and quality of references appropriate? Yes.

15. Is the amount and quality of supplementary material appropriate? NA

Technical corrections

Line 48: should read "greater than simulated under an equilibrium and climate warming

scenarios"? Line 71: NECB appears for the first time here but is defined at lines 162-163. Line 103: the "alternative hypotheses" are not clearly exposed and should appear here. Line 112-114: should be in the Discussion or Conclusion section. Line 117: same comment as SC1 Line 125: should read "Dunette et al. (2014)" Line 125-127: the sample resolution of the core results from the chronology based on 14C dates. I suggest to reorder the sentence. Line 129: should read "Dunette et al. (2014)" Line 160: autotrophic respiration is accounting in NPP yet. Line 163: how fire emissions are calculated in the model? Line 234: what is STATSGO? Line 252: should read "Figure 2" instead of "Figure 1". Line 275: should read "Kelly et al. (2016)". Line 275: should read "Together, this work and ours". Line 280: it is not clear what the equilibrium scenario is doing here. Line 286: can you justify the threshold of 1 Mg C ha-1? Line 296: should read "stand-replacing". Line 303: "lower" compared with equilibrium or paleo-informed scenario? Line 301: "As expected" refers to a hypothesis? I think you should present this hypothesis in the introduction. Line 301-305: you mention the equilibrium scenario in your comparison and refer to the Figure 3, but values for the equilibrium scenario don't appear in this figure.

Finally, I recognize the great potential of this paper and the important gap it helps to fill in the carbon cycling-related fire history knowledge. I am happy to see that such research is unfolding and I advise the authors to consider previous comments to improve their manuscript.

---

## Author Response (AR2)

Final manuscript corrections

We thank you for acceptance of our manuscript and we have made the changes to the final uploaded word document of the text (added the links to the Dryad repository and the Daycent documentation).

Thank you!

Tara Hudiburg, Philip Higuera, and Jeff Hicke
* * *
Relevant changes made in the manuscript per the reviews:

1. Climate scenario has been removed as a primary objective/hypothesis. The objectives of the study have been clarified and the modeling goals. We also discuss the modeling limitations given the lack of paleoclimate data.
2. Many portions of the text regarding methods, the site description, the model description, and the vegetation history has been revised and clarified (see specific comments).
3. The figures have been revised per the recommendations.
4. Many other changes regarding typos, citations, and wording have also been made per the recommendations.

Response to Editor comments

Editor comment 1 (rev 1 comment): I ask you to consider using existing AR4 or AR5 climate change scenarios to apply them to DayCent for your study region because these climate scenarios provide physically consistent climate variables for a 2-degree warming. Otherwise, the error propagation is too high and your results can be biased.

Editor Comment: Please make sure that this approach is thoroughly explained in the methods section. Also explain, why you cannot derive such type of information from your climate forcing data.

Author response: We agree that using climate forcing data that includes the other variables (like precipitation) would be a better way to test the impact of climate (rather than just warming). However, because our prescribed fire events are decoupled from climate in the model simulations, we chose not to pursue downscaled climate datasets with more physically constant variables as they would not influence the fire events (in the model). Finally, as requested by the editor, we have decided to go with option (2) advised by Rev 1 and eliminate the climate warming scenario from our hypotheses.

        In terms of other abiotic influences (precipitation and radiation), we agree they are important, but again, we do not and cannot easily acquire paleoclimate data for this watershed, making these impacts beyond the capability of the current study. Per the request, we have clarified this in the manuscript and discussed the limitations of the climate forcing data.

3. Net ecosystem responses cannot be derived from simulating fire pattern alone. Please re-
consider your response and revise your manuscript as demanded by reviewer 1.

Author response: We agree that net ecosystem response cannot be derived from simulating fire
pattern alone. We utilize a comprehensive, mechanistic, biogeochemical model (DayCent) that
includes the important processes that affect ecosystem response (vegetation, climate,
disturbance, plant growth, decomposition, etc) because of this reason. Per option 2 suggested by
Rev 1, we will "…explicitly present this study as a first-step modelling approach integrating only
the fire regime information and therefore only testing it" and remove the third hypothesis related
to climate. We will also discuss the limitations of the study regarding the climate forcing data.

4. Reviewer 1 has offered you two options for improving your manuscript. Please reconsider to
take one of the options to allow this manuscript getting published.

Author Response: As suggested by the editor, we are choosing option 2 (remove climate
scenario) as suggested by the reviewer and including text about the limitations of our climate
forcing data. In the discussion, we note the impact that 2 °C of warming in the model has on
plant growth and decomposition, relative to the changes from fire themselves. This sensitivity
analysis provides some coarse context for interpreting the magnitude of change from fire
activity, without implying that we have simulated past climate or coupled climate-fire-ecosystem
dynamics.

Editor comment 2: Reviewer 2:

1. Provide the information demanded by the reviewer in the manuscript text, accordingly.

Cf. Reviewer 2: Materials and Methods: L165 What exactly is the size of the simulated area?
Are fires  spatially-explicit? Or just based on random selection of cells? Perhaps a few word on
this.

Response: We have edited the text per Rev 2's requests, specifically where more information is
necessary.

2. Explore all available options for validating also vegetation composition or productivity as
demanded by reviewer  2: "This removes the necessity to do the paleo-informed, but nevertheless
paleodata comparison is necessary as a validation step" and describe it in the manuscript.

Author response: We have addressed this issue in the text. Specifically, we have clarified that the
vegetation composition has not changed and cited this information. There has not been any
dominant vegetation changes at this site for the study record. Also, we compare/evaluate our
productivity numbers with the only values available to us. We have also clarified this in the text.

Editor comment: In addition to these changes that need to be taken into account in the revision of
the current manuscript, all other changes demanded by the reviewers need to be considered. You
have announced that these changes were or will be conducted in the revised manuscript. These
changes will be essential.

Author response: We have edited the text and made the changes as requested and outlined in our
response.

Response to SC1

We thank the reviewer for thoughtful and helpful comments and have addressed many of the
suggestions (see specific replies below).

- Page 2, line 48: we suggest changing the word "great" to "greater" since it is followed by the word "than" and in comparison, certain adjectives such as great should get an "er" or "est"
at the end.

Response: This sentence has been removed.

- Page 4, line 83: we would change "significance influence of fire" to significant influence since
it makes more sense

Response: We have chosen to keep "of fire" as it more explicitly defines what we are referring to
(rather than climate).

For a better understanding and conception, we suggest the following: - Page 2, line 40: we would
find a definition of "C trajectories" helpful

Response: We have added the following clarification: "(i.e. future states or directions)" - Page 3,
line 61: it is somewhat unclear what the authors mean by pool sizes, we suggest that they
indicate which elements pool sizes they specifically mean (e.g. carbon or nitrogen or etc.,)

Response: Done.

- Page 3, line 71: it is not clear what is meant by Net Ecosystem Carbon Balance (NECB)

Response: Yes, this was unclear until the methods. Thank you for pointing this out. We have
now added text describing NECB (the balance between net forest carbon uptake and forest losses
through fire emissions).

- Page 4, line 86: the term "spin up" is confusing. We suggest that the authors try to explain and
clarify this term in a more understandable wording perhaps by defining this term with a simple
example before using it.

Response: We added the following sentence for clarification: "To initiate the model, C and N
pools need to develop, as they start from 'bare soil' with no vegetation; as vegetation grows the
modeled soil pools increase, and it takes hundreds to thousands of simulation years during this
"spin-up" period for the C and N pools to equilibrate.

- Page 5, line 139-141: "Day Cent" Is well described but already mentioned in section 2.1,
therefore we suggest the description should come earlier

Response: We switched the order of the sections so that the Model Description is now Methods
section 2.1 and the study site is section 2.2.

- Page 6, line 151-152: is L:N and lignin to nitrogen the same? It is not mentioned in the text

Response: Yes, we changed the L:N to lignin to nitrogen for consistency.

- Page 7, line 182: from our point of view, the "key difference" between the two fire types should
come at the beginning of the paragraph

Response: We moved "The key difference between the two fire types simulated is the associated
soil erosion" to the beginning (second sentence; line 181 now) of the paragraph.

- Page 8, line 208: timeframe CE, is that defined as common era?

Response: Yes, we added "common era" in parentheses.

- Page 8, lines 211-219: we think the explanation of different scenarios can be expressed in a
more precise and separated way. The description of additional scenarios make it difficult to
understand and follow the subject since they're told altogether. Perhaps by separating the
scenarios and explaining each of them on an independent paragraph, the concept can be easier to
follow. The use of that many brackets makes it more confusing than helping anything.

Response: We agree the descriptions were confusing. The text has been separated in to distinct
paragraphs with more explanation of each scenario.

- Page 9, line 248: isn't the data fitted? Not surprising that it is "broadly in agreement"

Response: Fire occurrence is "fitted", but not C losses. We include the comparison to indicate
that DayCent is capable (some models are not) of replicating the expected C emissions from fire
in this region.

- Page 13, line 360-365: very long and complicated sentence. We would suggest making more
than one sentence out of it for a better understanding

Response: This text has been changed (and edited).

- Page 13, line 369: the word "woody pool" should be clarified

Response: Done.

- Page 14, line 383 & 388: are "ecosystem states" and "biogeochemical states" the same? Here
we would need simplification or a better definition

Response: We are using them interchangeably, but decided to just use biogeochemical states.

Concerning the figures: - Implement results in Table 1

Response: We think providing the results in Table 1 would be repetitive, and thus unnecessary.

- Figure 1: For a better visual understanding, it would be nice to have at least two different colors
for the different types of fire. Also, different symbols could be used. The spacing between the
line is very big and could be better used. It would be sufficient to have only one legend as it is
the same, and we can read the word "high severity fire" four times in a small figure. That could
be simplified.

Response: We changed the fire severities to two different symbols (open vs closed) and now use
only one legend as well as making the symbols larger.

- Figure 2: It is too confusing that the grey Equilibrium line and the yellow Equilibrium + 2
degrees have the same value on the y-axis but it's not shown.

Response: We have removed the warming scenario from the figure.

- Figure 1, 2 and 4: In the text the time data is in CE. In the Figures time data Cal BP is used. We
would suggest to only use one time specification.

Response: Generally, tree-ring records that extend back several centuries (e.g., the tree-ring
inferred fire date at Chickaree Lake), are reported in years CE, while lake-sediment records,
which extend back thousands of years, are reported in years BP (to avoid negative values, prior
to 0 CE). We understand how this can be confusing, so we added years BP to the few places in
the text where we refer to year CE.

Reviewer 1 comment

First of all, even though the authors refer to past published studies, they should present or
document the reconstructed response of vegetation (changes or not) the site recorded at least with
the same level of information as for the fire reconstruction they provide.

Response: The pollen record at this site indicates the dominance of subalpine forest taxa
(lodgepole pine) for the duration of the record presented here, which is consistent with other
regional records (and therefore we so not vary the vegetation over time). We have clarified this
in the text. To support this statement, we provide the citation to the original paper with the pollen
record, as well as other studies from the region: Caffrey and Doerner 2012, Dunnette et al. 2014,
Higuera et al. 2014.

Secondly, and most importantly, I wonder why authors have used only the same fixed 30-year
time series for climate data whatever the time frame simulated over the last 4500 years BP
instead of using past climate simulations from GCM or ESM whose many have Holocene
climate as well as Future climate runs…. whereas several studies have documented and
discussed about the potential counter-effect of precipitation increase in compensating the effect
of temperature increase on fire occurrences and spread….

Response: We agree that using paleo and/or future climate scenarios would be very interesting
and useful. However, in this paper we are purposefully isolating the potential impacts of fire-
regime variability. Our intent is not to replicate the exact dynamics that occurred at Chickaree
Lake; rather, we are using DayCent as a tool to test alternative hypotheses and using the fire
history of Chickaree Lake as an example of realistic variability in fire activity. In DayCent, we
thus prescribe when fire events occur, which automatically decouples the fire events from
climate from a modeling point of view. Even if we had a perfect paleoclimate data, few (if any)
models would be capable of replicating the Chickaree Lake record, which would turn the paper
into a model development project. Additionally, we also prescribe the erosion events associated
with fires, again decoupling them from precipitation events.

This would have prevent authors from saying that fires and climate are disconnected which is
absolutely not true, or at least need to be tested for each ecosystem studied. Moreover, instead of
just increasing the 30-year time series temperature by 2°C, they could have used the full climate
time series for the 21st century simulated by the same climate or earth models that provided the
Holocene runs. They even could have tested different IPCC scenarios and their impact of the
NECB. The use of climate model data would have provided precipitation time series as well,
whose changes could also have impacted soil nutrient (and C) leaching. Indeed, it is easy to show
that fire regime change outweighs climate change when such climate change may be unrealistic
or only taken into account through temperature increase whereas several studies have
documented and discussed about the potential counter-effect of precipitation increase in
compensating the effect of temperature increase on fire occurrences and spread.

Response: We certainly do not believe that climate and fire are disconnected, and much of our
own work explores fire-climate relationships in these and other ecosystems. To clarify this, we
added a note in the study area description, briefly specifying the nature of fire-climate relationships in regional subalpine forests and citing a key reference. In DayCent, the only
impact of using forced climate (with the forced fire and erosion events) would be the feedbacks
to plant growth, which would increase or decrease the biomass available to burn given certain
climate conditions. This is why we implemented the simple warming scenario: to see if/how our
results would differ when biomass accumulation rates were higher (due to warmer temperatures).
Our results indicate that the impacts of climate, as reflected by plant growth, is insignificant
compared to the disturbance impacts in the model. However, we agree that this is not a good way
to test the impact of climate on C cycling over time at this site and because this was not our
intent, we have removed the warming scenario from study design in manuscript. We refer to the
impacts of a 2 °C warming simply as a sensitivity analysis within the context of the DayCent
model only, and not as a scenario representing coupled climate-fire-ecosystem dynamics.

Finally, because the charcoal record indicates when fire events occur, incorporating a
paleoclimate record at the daily timestep and for a single location in the Rocky Mountains would
likely add significant uncertainty, in both the precipitation regime and certainly if fire was
"dynamic" and occurred in response to simulated climate.

Reviewer: It is even more important in the studied system as authors suggested and used two
types of high severity fires: those with and those without erosion. Stand-replacing fires (95%
mortality) are not really severe fire if post-fire regeneration is occurring in the next following
years from naturally adapted species. Fire severity would rather refer to the difficulty of post-
regeneration encountered in special cases. Stand-replacing fires are usually very intense and fuel
consumption includes all the litter and humus layers, leaving the mineral soil exposed. So, if
erosion in the burned watershed occurs (towards the lacustrine receptacle), it is performed during
(heavy) rainfall events. Therefore, this is another argument to show that it would have been
valuable to use past simulated precipitation over the last 4500 years BP, in order to test if rainfall
(even as mean annual rainfall) changes could have occurred contemporaneously to erosive events
just after some fires as compared to others.

Response: In western North America, subalpine forests like our study area are classified as
"high-severity fire regimes," where "severity" refers to the immediate impacts of a fire on the
ecosystem, often measured (directly or indirectly) by the amount of vegetation killed. In most
cases, post-fire regeneration in subalpine forests does indeed start in the year immediately
following fire, but we consider this an ecosystem response. While we appreciate the
shortcomings of the concept of "fire severity," this is the standard terminology used, and we have
added some references to support this use (i.e., Keeley 2009, Int. Journal of Wildland Fire). We
simulated consumption of litter and humus layers in DayCent. In fact, the fires were
parameterized to consume (combust) the forest biomass pools given known combustion
coefficients for these types of forests (which includes 99% removal of the litter layer). With
respect to climate forcing, again, we are forcing the erosion events to occur regardless of
precipitation, based on the reconstructed fire history record. It would be ideal to test if the
erosion events occurred with large precipitation events/years, but this is beyond the scope of this
study.

Moreover, authors provide no information on the vegetation compartment modeled except the
Net Ecosystem

Production for outputs, so we have no idea about which plant types are used for this site nor why
30cm deep was chosen as the targeted depth to analyze the site response. Finally, in the current
version, except from NEP, we have not idea about the effect of vegetation change in terms of
composition nor structure through time, we cannot see the direct as well as indirect effects of
climate change on vegetation nor climate on fire as climate dataset was fixed and repeated along
the 4500 years BP, even though fire ignition and fire spread conditions may have been more or
less favorable.

Response: Our purpose in this study is not to predict the effects of climate (or fire) on vegetation
change over time (or the effects of $CO_2$ or nitrogen deposition, etc). The study site description
includes a description of the known vegetation cover and based on the previously published
pollen record from this site and others, we are confident that this general forest type did not
change over the duration of our record (as noted above). DayCent (and most biogeochemical
models) can only model soil C dynamics to a depth of 30 cm, primarily because this is the most
active zone. The vegetation history has been more thoroughly described in the text, with
additional references for support.

For all these reasons I see two options that require to modify the manuscript:

Option 1: to do the modelling experiment exercise once again but using climate data that
represent the studied Holocene period for the first part and the 21st century for the second part.
Even though climate data come from GCM and are not perfect, they will still be better than
present-day ones applied to past and/or future periods, especially if climate is tested and its
relative impact compared to that of fire regime variability. In parallel to temperature and
precipitation datasets, authors should explain how they deal with air $CO_2$ concentration as it
should have been modified from 280 ppmv until 1750 to the historical recorded concentration
until nowadays, and for the Future, at least a mean $CO_2$ increase should be used if authors do not
want to test several RCP scenarios. By keeping the $CO_2$ at a fixed concentration could still be
acceptable but once more, as they are tracking C pools, I think that the atmospheric C input
should be taken into account.

Response: This is beyond the scope of this study and we are concerned that this activity would
introduce large amounts of uncertainty (given modeling limitations) rather than actually
clarifying our results. Again, our purpose here was not replicate the exact Holocene dynamics of
this site (although we agree this is an important next step/project).

Option 2: keep the modelling experiment in the current version but authors need at least to
remove the third objective as climate has not been properly taken into account as compared to
the fire regime factor. In such case, they should explicitly present this study as a first-step
modelling approach integrating only the fire regime information and therefore only testing it. All
sentences related to climate effect should be modified in order to rather present or discuss limit
of non-using proper climate data. This would better fit with the balanced way results must be
discussed. In such a case, the first two objectives are still OK. Results and conclusions should be fairly presented without omitting that the climate data used may be a limit to the interpretations
done.

Response: We agree the climate objective should not be a 'main focus' or main objective of the
paper. We have removed the third climate objective.

Otherwise, I found pertinent the improvements suggested in the M.W.I. Schmidt's comment
posted for improvement definitions, more detailed explanations and improvement in figure
quality so I encourage the authors to take them into account. They will facilitate the reading of
the manuscript for people not fully familiar with model requirements and functioning such as the
need of a spinup period, the use of several pools or compartments… If supplementary material is
allowed I suggest to add such information there, even with a scheme presenting how the
DayCent model works.

Response: We have addressed and utilized many of the comments from Schmidt. DayCent has
excellent documentation online (powerpoints, step by step instructions, publication lists;
http://www.nrel.colostate.edu/projects/daycent-downloads.html). If allowed we will include the
link in the manuscript. We will also post our model input and output on the Dryad repository (not
allowed until manuscript is published).

Response to Rev 2

Also, aside from discussing the biogeochemical elements, it could be interesting to also compare
some of the ecological attributes like age distribution of forest stands between the paleoinformed
and equilibrium approaches. Clearly the distribution of ages will be quite different, which could
have implications if eventually model simulations become a tool for forest management
guidelines aiming at sustainability of ecological services.

Response: We agree examining other ecological attributes would be interesting. The reviewer
has hit on a frustrating problem in the ecosystem modeling world, especially as it pertains to
providing useful tools for management. Unfortunately, DayCent (and most BGC models) do not
model age distributions or forest structural changes, as there are no 'trees' explicitly modeled. To
model individual trees, one needs to use forest landscape/succession models, which either lack
the biogeochemistry or operate a spatial scales much too large for this project (like LPJ as
suggested below). We also believe the soil model in released/validated versions of LPJ is
insufficient for this project.

Specific comments

Introduction:

L87-93 Would this rather illustrate that many models that perform a spin-up period lack a
validation of their simulated biochemical cycle?

Response: Spin-up is a necessary step given the need to reach steady state (and have an
ecosystem with 'states' to model). We agree that it is/has been difficult to validate spin-up and
spin-up as rather been used to reflect realistic 'steady states'. With the advent of more paleo data,
more spin up validation could be done.

Typically, the period after spin-up (what we refer to as equilibrium in this study) is validated
against current ecosystem states, given information available. For DayCent, validation of the
biogeochemical cycling has been performed in 100s of studies for 1000s of data points,
originally published as the CENTURY model (Parton et al. 1983) with many publications in all
types of terrestrial ecosystems since then.

Materials and Methods:

L165 What exactly is the size of the simulated area? Are fires spatially-explicit? Or just based on
random selection of cells? Perhaps a few word on this.

Response: This was a 'point' simulation (size is not explicitly modeled) for a single study site.
The simulation represents the watershed (c. 30 hectares) that would be affected in a high-severity
fire with erosion. The fire is spatially-explicit to the single point, as there are no other
points/grids. We have clarified that this is a point simulation in the text.

L176 So climate and radiation are constant. This may be problematic because in the eventuality
that climate was different during the late-Holocene, as compared to the Anthropocene, likely the
simulation will be misleading the productivity levels. So I guess this is another argument for doing the +2C and -2C simulation experiments (L217-224). Not using paleoclimatic simulation
is an important weakness of this study and I would recommend that authors put more emphasis
on the importance of this temperature sensitivity analysis. However, they should note that
temperature is not the only driver of NPP; radiation and precipitation are also important.

Response: As pointed out by Rev. 1, climate impacts are not (and should not be) a main focus of
the study. We agree that using paleo and/or future climate scenarios would be very interesting
and useful. However, in this paper we are purposefully isolating the potential impacts of fire-
regime variability. Our intent is not to replicate the exact dynamics that occurred at Chickaree
Lake; rather, we are using DayCent as a tool to test alternative hypotheses and using the fire
history of Chickaree Lake as an example of realistic variability in fire activity. In DayCent, we
thus prescribe when fire events occur, which automatically decouples the fire events from
climate from a modeling point of view. Even if we had a perfect paleoclimate data, few (if any)
models would be capable of replicating the Chickaree Lake record, which would turn the paper
into a model development project.

In terms of the temperature sensitivity, we show that net C balance is not sensitive to temperature
relative to the impacts of disturbance, and this was really just a check on what we already know
about climate vs. disturbance impacts (as pointed out by Rev. 3). In terms of other abiotic
influences (precipitation and radiation), we agree they are important but again, we do not and
cannot easily acquire paleoclimate data for this watershed, making these impacts beyond the
capability of the current study. We include the temperature sensitivity results as a simple test on
the model, although they are no longer a main focus.

L182-185 More details are needed in regard to the validation dataset. What kind of datasets are
these observations? How were they derived? Why select these over others? What do you mean
by 'similar-aged'?

Response: There are very few observations (carbon, nitrogen pools, NPP, etc) for old (200+ yr)
stands of lodgepole pine in the Rocky Mountains. The studies were chosen given that they had
reported variables the most similar to our model output, were for the same species or taxa, and
were in similar environmental/climate conditions. 'Similar-aged' means the same forest age. We
do not consider these comparisons with reported observations a robust validation dataset; rather,
this is the only means of validating some of the model output. We have clarified this in the
manuscript.

Results and Discussion:

L241 What are the plus and minus signs for? Standard deviation or confidence intervals? What is
the sample size? Area under analysis? Seems that crucial details are missing.

Response: The plus/minus signs are the standard deviation for the range of bulk density and soil
organic matter percent reported for the dominant soil type that occurs in the Chickaree
watershed. Soil carbon can be derived from STATSGO data (US federal database). This has also
been clarified in the manuscript.

L274-278 This statement about disturbance free or intensified disturbance periods is partly false,
because DGVMs now have the capacity to run fire dynamics using paleoclimate simulations that
feed into a dynamic fire behaviour and growth model (e.g., LPJ-LMfire). This removes the
necessity to do the paleo-informed, but nevertheless paleodata comparison is necessary as a
validation step.

Response: Yes, there are models (and not just DGVMs) with prognostic fire, so yes there could
be predictions of disturbance-free periods (and more intense ones). However, there are few
models that actually duplicate known records of ignitions, burn area, and most importantly for
this study, carbon combustion; we are unaware of any models with reasonable accuracy at the
point scale. We chose DayCent because of its proven ability to predict above and belowground C
dynamics at daily to millennial scales. We are also unaware of downscaled paleoclimate
simulations that are 'readily available' at high spatial resolutions for this region.

L294-298 This is not really new and has been known for decades. The impact of fire versus
vegetation is quite obvious considering that fire has the potential to exclude treed vegetation
from landscapes despite generally improving growth conditions with warming and CO2

Response: Yes, we agree and have changed the wording to reflect that our results confirm what
has been known about the impacts of individual fire events, for decades. The 'new' information
has more to do with the impacts of the varying timing/sequence and severity of events over
centuries to millennia. Certainly, any given fire will outweigh climate impacts in early post-fire
recovery. Here, we show that the timing and severity of events over centennial and millennial
scales strongly influences the state and trajectory of biogeochemical properties.

L343 "the lack of paleoclimate data" : this is an important weakness of this study. A few
sentences about this is needed here to help readers unfamiliar with this issue to understand what
is meant by 'paleoclimate data'.

Response: We agree that not using paleoclimate data is an important limitation of our study, and
our intention in this portion of the text is to clearly frame our results in this context. Although
paleoclimate proxies exist for other regions in Colorado, for example in the form of lake-level
reconstructions and oxygen isotope records, these records are far from the detailed climate
information needed to drive DayCent. Thus, utilizing paleoclimate proxies to develop climate
drivers for DayCent is a project in itself. For example, it involves developing methodologies to
downscale paleoclimate proxies in space (to the elevation and location of Chickaree Lake), in
time (to daily value), and to the specific metrics required by DayCent (e.g., from a relative
moisture proxy to daily precipitation). We added text to further clarify this limitation and why
this was not done in this study.

Figures: Figure 4 This figure is not obvious to read. Perhaps put on separate panels.

Response: Thank you for the comment. We have separated the panels.

Response to Rev. 3

General comments…Globally, the text is clearly written, the scientific context and knowledge gaps are clearly exposed as the problematic and the general hypothesis. Also, the questions addressed here are very pertinent. That said, I advise the authors to follow previous comments and advises from SC1, RC1 and RC2. Moreover, a more deeper review of fire ecology with respect to carbon cycling could: i) help to better understand the choice of DayCent for this study; ii) bring a more critical interpretation/discussion of the processes you mentioned (line 99-100) linked in DayCent model and improve the interpretation and discussion of the results.

Response: We thank you for the careful review and suggestions. Please see our specific comments below for our planned improvements.

I also noted several improvement possibilities (see also Technical corrections): 1/ Structure: Mixing results and discussion is sometimes confusing (especially for section 3.4). Because section 3.1 to 3.3 are not full discussions but rather descriptions and comparisons between your model estimates with values of other studies, it should not will be difficult to separate results and discussion. For example, discussion could contain a section on the limits, a section with the implications for projecting future ecosystem states and another for research development needs.

Response: We will consider revising the structure to separate the results and discussion based on the final revised manuscript. Because of what we address from the first 3 reviewer comments, the structure and text has changed enough that doing these structural improvements may no longer be straight forward.

2/Hypotheses: Based on Kelly et al. (2016), the general hypothesis assuming forest carbon budget modeling would be different between equilibrium runs and paleo-informed runs is explicit. Nevertheless, the alternative hypotheses that you mentioned (line 103) and results that were "expected" (line 301) are not explicitly described. You could add these hypotheses in the introduction.

Response: Thank you for pointing this out. We have changed the introduction to more explicitly state theses hypotheses.

3/ Model parameterization: According to SC1, DayCent is quite well described. Unfortunately, I was not able to access the model input and parameterization file. While is it clear that you informed the model with paleo-fire reconstruction from Dunette et al. (2014), it is less clear what you do with the vegetation data. You wrote that you "pair a paleoecological record of vegetation and wildfire activity" (line 98) and that DayCent requires input of vegetation cover (line 145), but no information is provided on vegetation in section 2.3. It would be important to get more details.

Response: The comments here is in agreement with Rev 2, and we realize details need to be expanded regarding the simulations. We will add the details (note that the 'vegetation' did not change at this site per the record). We plan to post the DayCent input files on Dryad, however, this is not allowed until publication.

Specific comments

10. Is the overall presentation well structured and clear? Yes, but could be improved (see
General comments). 11. Is the language fluent and precise? Yes. 12. Are mathematical formulae,
symbols, abbreviations, and units correctly defined and used? Yes, but see SC1 comments for
[date] CE.

Response: We have clarified this.

13. Should any parts of the paper (text, formulae, figures, tables) be clarified, reduced, combined,
or eliminated? Yes. Values for equilibrium scenario should appear in Figure 3 or equilibrium
scenario should be removed in lines 301-305. As the Chickaree Lake watershed is the object of
this study, some characteristics such as the watershed size and topography (slope characteristics)
could be mentioned. Moreover, you defined 8 partial paleo-informed scenarios but only 4 are
represented in Figure 1. To facilitate the reading, I suggest to represent all partial paleo-informed
scenarios in Figure 1 or you can specify that you show only 4 on the 8 scenarios in the figure
caption.

Response: We improved the figures and text as suggested.

14. Are the number and quality of references appropriate? Yes.

Technical corrections Line48: should read "greater than simulated under an equilibrium and
climate warming scenarios"?

Response: This text has been removed from the abstract.

Line 71: NECB appears for the first time here but is defined at lines 162 163.

Response: This has been addressed.

Line 103: the "alternative hypotheses" are not clearly exposed and should appear here.

Response: As noted above, we have revised the hypotheses.

Line 112-114: should be in the Discussion or Conclusion section.

Response: This text has been removed (it was basically repeated in the discussion).

Line 117: same comment as SC1 Line 125: should read "Dunette et al. (2014)"

Line 125-127: the sample resolution of the core results from the chronology based on 14C dates.
I suggest to reorder the sentence.

Line 129: should read "Dunette et al. (2014)"

Line 160: autotrophic respiration is accounting in NPP yet.

Response: We have revised based on the suggestions above.

Line163: how fire emissions are calculated in the model?

Response: We added text to clarify this. Basically, the fire is parameterized by pool (woody,
litter, coarse wood, live or dead C) to combust a fraction of each pool based on the fire
'severity'.

Line234: what is STATSGO?

Response: The definition and a general description of the database will be added (USDA soils
database from the Natural Resource Conservation Service).

Line252: should read "Figure2" instead of "Figure1".

Line275: should read "Kelly et al. (2016)". Line275: should read "Together, this work and ours".

Line 280: it is not clear what the equilibrium scenario is doing here.

Line 286: can you justify the threshold of 1 Mg C ha-1?

Response: Again, thank you for the careful reading! We addressed the corrections, clarified what
equilibrium is doing and, yes, we can justify the threshold based on previous work and what we
consider to be stable soil C.

Line 296: should read "stand-replacing".

Line 303: "lower" compared with equilibrium or paleo-informed scenario?

Line 301: "As expected" refers to a hypothesis? I think you should present this hypothesis in the
introduction.

Line 301-305: you mention the equilibrium scenario in your comparison and refer to the Figure
3, but values for the equilibrium scenario don't appear in this figure.

Response: As noted above, we changed the introduction as suggested and the figure is comparing
the final values to equilibrium (they are deltas).

Finally, I recognize the great potential of this paper and the important gap it helps to fill in the
carbon cycling-related fire history knowledge. I am happy to see that such research is unfolding
and I advise the authors to consider previous comments to improve their manuscript.

Response: Thank you!

[revised manuscript text omitted]